# Perceived Control and Work-Related Stress Mediate the Effects of Grit on Depression among Employees

**DOI:** 10.3390/brainsci13010009

**Published:** 2022-12-20

**Authors:** Sra Jung, Young Chul Shin, Kang-Seob Oh, Dong-Won Shin, Eun Soo Kim, Mi Yeon Lee, Sung Joon Cho, Sang-Won Jeon

**Affiliations:** 1Department of Psychiatry, Kangbuk Samsung Hospital, Sungkyunkwan University School of Medicine, 29 Saemunan-ro, Jongno-gu, Seoul 03181, Republic of Korea; 2Workplace Mental Health Institute, Kangbuk Samsung Hospital, Seoul 04514, Republic of Korea; 3Department of Biostatistics, Kangbuk Samsung Hospital, Sungkyunkwan University School of Medicine, Seoul 04514, Republic of Korea

**Keywords:** workplace mental health, depression, grit, occupational stress, perceived control

## Abstract

We developed and evaluated an interpersonal model of depression in employees, where passion and perseverance affect occupational stress and perceived control, which in turn affect risk of depression. The participants were employees of 18 private companies and local government organizations in Korea aged 19 to 65 years. A total of 11,422 participants completed questionnaires including the Korean version of the Grit scale, the occupational stress scale, the perceived control subscale in the perceived stress scale, and the Center for Epidemiologic Studies Depression Scale. Mediation analysis was performed to determine relationships among trait-level passion and perseverance, work-related stress, perceived control, and depression. Passion and perseverance preceded depression in employees. Higher occupational stress and lower perceived control mediated the association between passion and depression, and between perseverance and depression. Passion and perseverance exert preventive effects on depression by decreasing workplace stress and elevating perceived control. Future studies should investigate the effects of psychological characteristics on the development of depression in employees.

## 1. Introduction

The prevalence of major depression in South Korea is gradually increasing. Major depression is characterized by repeated depressive episodes and the inability to function normally during each episode. According to the Global Burden of Disease statistics [1] released in 2022, depression was among the top ten causes of disability worldwide. In the workplace, depression can negatively influence productivity through increased absenteeism and presenteeism [2]. In South Korea, the mean annual cost per person for absenteeism associated with depression was $181, and the mean presenteeism cost per person associated with depression was $6066 [2]. As workplace depression incurs a significant social cost, public interest in depression among employees continues to grow.

Grit is defined as passion and perseverance to meet long-term goals [3], and also as the tendency to sustain interest and commitment to achieving worthy goals even if it requires sacrifice, struggle, and suffering [4]. Passion is a strong feeling toward a personally important value that leads individuals to consistently desire to engage in the important value [5]. Passion combined with persistence increases immersion in activities, and increases concentration and pursuit that allows individuals to continue their efforts [6], thus acting as an important factor in achieving goals. Perseverance acts to sustain goal-seeking behavior, while passion acts as the energy to focus on achieving goals [7]. Perseverance of grit overlaps with the concept of conscientiousness among the “Big Five” personality traits [8,9]. Conscientiousness implies thorough, careful, reliable, organized, industrial, and self-controlled character. Despite some similarities in characteristics, perseverance differs from conscientiousness. Although conscientious individuals complete tasks in one breath, individuals with high perseverance make steady and persistent efforts to achieve goals over long periods [10].

Since grit has been considered as an important factor in coping with mental health pressure, several previous studies have examined the relationship between grit and various psychiatric symptoms such as depression, anxiety, insomnia, and suicidal thoughts. In a cross-sectional study on university students in Thailand [11], gritty subjects showed lower levels of depression and anxiety. A study on emerging adults between the ages of 18 and 25 in the United States [12] and a study on Turkish university students [13] also showed that depression and anxiety were less likely to occur in subjects with high grit. Similarly, a study on the general population in Sweden during the COVID-19 pandemic [14] showed that people with higher grit reported fewer symptoms of depression, anxiety, and insomnia during the pandemic era. In addition, a previous study [15] showed that among individuals who experienced trauma, those with high grit experienced less posttraumatic stress symptoms and complained less about suicidal ideation. There are preceding studies which show that grit plays an important role not only in various psychiatric symptoms, but also in various psychological factors that have recently been in the limelight, such as positive psychological factors, quality of life, or happiness. Previous studies have shown that gritty individuals feel happier and have more life satisfaction [16], and people with high grit have high self-efficacy [17], high self-esteem [18], low perceived stress [19], and are more emotionally stable during stressful or negative life events [20]. Combining these previous studies, gritty individuals who stay passionate and persevere against frustration may interpret failure more optimistically and be less prone to negative emotions and depressive episodes. Although the relationship between grit and depression in individuals is generally consistent, the mechanisms by which grit protects against depression have not been fully elucidated. Given that individuals with high grit experience fewer psychiatric symptoms and have more positive psychological factors, it is important to study grit in the workplace for employees who are inevitably exposed to stressful situations and need to maintain their mental health and productivity. However, the number of studies examining the effect of grit on employees is insufficient.

In previous studies, grit was regarded as a personality trait and considered an individual’s characteristic that is not easily changed in normal conditions in adults [9]. However, a previous study [21] of 2321 pairs of twins revealed that grit’s heritability was only 20% for passion and 37% for persistence, indicating that environmental factors played a more significant role than genetic factors in the formation of grit. From a neuroscientific view, passion and persistence are mainly formed in early life. Executive functioning is the ability to maintain focus on a given task rather than responding to new stimuli, and underlies goal formation, planning, carrying out goal-directed plans, and effective performance [22]. These descriptions of executive functioning overlap significantly with the definitions of passion and persistence. Executive functioning is known to mainly involve the frontal cortex, and the development of the frontal cortex is known to proceed until late adolescence [23,24]. It can be expected that grit in context with executive functioning is therefore formed through interactions with the surrounding environment in childhood and adolescence.

Hence, in this study we investigate the relationship between grit already formed as an adult and depression in the workplace, and whether occupational stress and perceived stress underlie the relationship between grit and depression. Our hypotheses were as follows:Both types of grit, passion, and perseverance are inversely related to occupational stress and depression, and positively related to perceived control in employees.Higher levels of occupational stress are associated with more severe depression and greater perceived control has a protective effect against depression in employees.Occupational stress and perceived control mediate the relationship between grit and depression in the workplace.

## 2. Materials and Methods

### 2.1. Participants

The study participants were male and female workers who underwent workplace mental health screening at the Workplace Mental Health Institute of Kangbuk Samsung Hospital, Seoul, Republic of Korea, between April 2020 and March 2022. This study included participants over 19 at the beginning of the study in order to involve legal adults, and participants under 65 to prevent the effects of cognitive decline. The subjects were office or manufacturing-facility workers at one of the country’s 18 public institutions or large companies who voluntarily participated in mental health examinations upon invitation from their companies. Among the initial 12,344 respondents, only 11,422 (7750 men and 3672 women) participants who completed questionnaires and sociodemographic information were included in the final analysis.

The participants included employees of 2 companies (126 participants) in the field of manufacturing; 2 companies (1668 participants) in the field of transportation and storage; 5 companies (3575 participants) in the field of financial and insurance activities; 1 public institution (903 participants) in the field of education; 1 company (960 participants) and 346 other participants in the field of human health and social work activities; 1 company (57 participants) in the field of arts, entertainment, and recreation; 1 company (2933 participants) in the field of other service activities; and 5 companies (1775 participants) in the field of public administration and defense, and compulsory social security [25].

All study procedures were approved by the Institutional Review Board of Kangbuk Samsung Hospital and adhered to the latest version of the Declaration of Helsinki and principles of Good Clinical Practice (Approval number: KBSMC 2022-03-046). The requirement for informed consent was waived because only de-identified data routinely collected during health screening visits were used.

### 2.2. Clinical Assessments

We collected socio-demographic data including age, gender, level of education (high school or below, college graduate, university graduate, master’s, and doctorate), and marital status (married, unmarried, other [divorced, widowed, and separated]). Job-related demographic factors were also collected, including duration of work at the current workplace (years), hours of work per week, and monthly earned income.

Grit was assessed with the eight-item Short Grit Scale [26], an eight-item self-reported questionnaire with responses scored on a five-point Likert scale ranging from 1 = not at all like me to 5 = very much like me. This scale focuses on two factors: passion (three items, Cronbach’s alpha: 0.585) and perseverance (five items, Cronbach’s alpha: 0.779). The Cronbach’s alpha of the eight-item Short Grit Scale was 0.794.

Occupational stress was assessed using the Short Form of the Korean Occupational Stress Scale-Short Form (KOSS-SF) [27], a 24-item self-reported questionnaire scored on a 4-point Likert scale ranging from 1 = strongly disagree, 2 = disagree, 3 = agree, and 4 = strongly agree. Higher scores indicate greater workplace stress. It is composed of seven work-related stress subscales: high job demands (four items, Cronbach’s alpha: 0.816), insufficient job control (four items, Cronbach’s alpha: 0.669), inadequate social support (three items, Cronbach’s alpha: 0.720), job insecurity (two items, Cronbach’s alpha: 0.748), organizational injustice (four items, Cronbach’s alpha: 0.804), lack of reward (three items, Cronbach’s alpha: 0.786), and discomfort in an organizational climate (four items, Cronbach’s alpha: 0.735). The KOSS has previously shown high internal consistency and validity. The Cronbach’s alpha for the Korean version of the KOSS was 0.895. We additionally measured occupational stress related to the physical environment using the Korean Occupational Stress Scale (KOSS) physical environment sub-scale (three items, Cronbach’s alpha: 0.560). When Korean occupational stress is expressed using the simple sum of subscale scores, the number of items in each subscale area of KOSS is not the same and therefore some area scores may be excessively weighted. In order to compensate for this problem, the following formula was used for the evaluation of the total Korean occupational stress [27].
$$\mathrm{Corrected}\;\mathrm{KOSS}\;\mathrm{subscale}\;\mathrm{score}=\frac{\mathrm{Actual}\;\mathrm{score}-\mathrm{Number}\;\mathrm{of}\;\mathrm{questions}}{\mathrm{Highest}\;\mathrm{possible}\;\mathrm{score}-\mathrm{Number}\;\mathrm{of}\;\mathrm{questions}}\times 100$$
$$\mathrm{KOSS}\;\mathrm{total}\;\mathrm{score}=\frac{\mathrm{Sum}\;\mathrm{of}\;7\;\mathrm{corrected}\;\mathrm{KOSS}\;\mathrm{subscale}\;\mathrm{scores}\;\mathrm{except}\;\mathrm{physical}\;\mathrm{environment}\;\mathrm{subscale}\;\mathrm{score}}{7}$$

Daily perceived control was measured using the positively worded items of the Korean version of the 10-item Perceived Stress Scale (PSS) [28], which measures the degree to which the respondent perceives some current aspect of the event to be controllable. It is a self-reported questionnaire with responses scored on a five-point Likert scale ranging from 1 = never to 5 = very often. The Cronbach’s alpha for the Korean version of the PSS was 0.813.

Depressive symptoms were assessed using the Korean version of the 20-item Center for Epidemiological Studies Depression (CES-D) scale [29]. It is a self-reported questionnaire with responses measured on a four-point Likert scale ranging from 0 to 3 points. Higher total score indicates more severe depression. Cronbach’s alpha for the Korean version of the CES-D was 0.765.

### 2.3. Statistical Analysis

We conducted descriptive analyses of the demographic characteristics and correlation analyses of the four variables. To test the first and second hypotheses, Pearson correlations were performed for the scales to assess the direct associations between grit and depression and to support the associations between occupational stress and perceived control with the other variables: grit and depression. 

The mediation model was analyzed using Model 4 in the PROCESS Macro in SPSS, developed by Hayes [30]. Bootstrapping is one of the most valid and powerful methods for testing multiple mediation models. Bootstrapping is a nonparametric resampling procedure which involves repeatedly sampling from the data set and estimating the indirect effect in each resampled data set. For the best test of mediation effects, bootstrapping was carried out with 5000 samples to measure indirect effects, and 95% confidence intervals were estimated. If the confidence interval included zero, it meant that there was no significant mediating (indirect) effect at the significance level of 5%. Mediation analyses were performed using two independent variables (passion and perseverance), one dependent variable (depression), and two mediators (occupational stress and perceived control). For each multiple mediation model, two specific indirect effects of grit on depression, one via occupational stress and the other via perceived control. A specific indirect effect represents the ability of occupational stress or perceived control to mediate the effect of grit on depression controlling for the other mediator. The total indirect effect of grit on depression summarizes the specific indirect effects. Additional mediation analyses between passion and perseverance with depression were performed with subscale scores of KOSS as mediators. Control variables such as age, sex, education, marital status, years of service, hours of work per week, and monthly earned income were introduced in the model as covariates. All analyses were performed using SPSS 28.0 for Windows.

## 3. Results

### 3.1. Demographics, Job Characteristics, and Clinical Characteristics of the Study Participants

Table 1 displays the demographic, job, and clinical characteristics of participants. The mean age of participants was 36.7 ± 9.4 years, and 7750 (67.9%) were male. CES-D scores showed significant positive correlations between age and hours of work per week and significant negative correlations between years of service and monthly earned income. Female gender, lower degree of education, and unstable marital status (e.g., divorced, widowed, or separated) were associated with higher CES-D scores.

### 3.2. Correlations

The Pearson correlations for each of the five variables (i.e., passion, perseverance, occupational stress, perceived control, and depression) are presented in Table 2. Passion was positively correlated with perseverance (r  =  0.495, *p*  <  0.001) and perceived control (r  =  0.325, *p*  <  0.001), and inversely associated with occupational stress (r  =  −0.378, *p*  <  0.001) and depression (r  =  −0.440, *p*  <  0.001). Perseverance was positively correlated with perceived control (r  =  0.520, *p*  <  0.001) and negatively associated with occupational stress (r  =  −0.430, *p*  <  0.001) and depression (r  =  −0.438, *p*  <  0.001). Occupational stress was negatively correlated with perceived control (r  =  −0.443, *p*  <  0.001) and positively associated with depression (r  =  0.620, *p*  <  0.001). Perceived control was negatively correlated with depression (r  =  −0.548, *p*  <  0.001). There were significant positive inter-correlations between scores of subscales of KOSS. Passion, perseverance, and depression were inversely associated with all subscale scores of KOSS (Appendix A).

### 3.3. Mediation Effect Testing

As age, sex, degree of education, marital status, years of service, hours of work per week, and monthly earned income were significantly associated with depression in the workplace, these variables were controlled in mediation effect evaluations. The PROCESS macro developed by Hayes [30] was used to determine the multiple mediating roles of occupational stress and perceived control in the relationship between grit (passion and perseverance) and depression among workers. Table 3 shows the mediation analysis results, while Figure 1 and Figure 2 show the mediation pathway models.

The significance of the mediating effects was tested using a bootstrap method. Table 4 shows the results of the bootstrap analysis. First, the total effects, direct effects, and total mediating effects of occupational stress and perceived control between passion with depression were −1.985, −0.870, and −1.115, respectively, that is, the total mediating effects accounted for 56.17% of the total effects. Specifically, the effect of the path passion→occupational stress→depression was −0.658, accounting for 33.16% of the total effects, and the effect of the path passion→perceived control→depression was −0.457, accounting for 23.01% of the total effects. Second, the total effects, direct effects, and total mediating effects of occupational stress and perceived control between perseverance with depression were −1.206, −0.254, and −0.952, respectively, that is, the total mediating effects accounted for 78.97% of the total effects. Specifically, the effect of the path perseverance→occupational stress→depression was −0.501, accounting for 41.57% of the total effects, and the effect of the path perseverance→perceived control→depression was −0.451, accounting for 37.39% of the total effects.

We further explored the mediating effects of the eight subtest scores of KOSS on the relationship between grit and depression. As seen in Appendix A, passion and perseverance both have significant effects on each mediator variable: difficult physical environment, high job demand, insufficient job control, inadequate social support, job insecurity, organizational injustice, lack of reward, and discomfort in occupational climate. As for the relationship between passion and depression, all subtest scores except for difficult physical environments have significant effects on depression. The mediating effects of all subtest scores except for difficult physical environment on the relationship between passion and depression are all significant, in the order of lack of reward, high job demand, job insecurity, discomfort in occupational climate, inadequate social support, organizational injustice, and insufficient job control (Appendix A). As for the relationship between perseverance and depression, all subtest scores except for difficult physical environments and insufficient job control have significant effects on depression. The mediating effects of all subtest scores, except for difficult physical environments and insufficient job control, on the relationship between perseverance and depression are all significant in the order of lack of reward, discomfort in occupational climate, job insecurity, high job demand, inadequate social support, and organizational injustice (Appendix A). In addition, the direct effects of passion (B = −0.619, SE = 0.023, *p* < 0.001) and perseverance (B = −1.092, SE = 0.037, *p* < 0.001) on depression remain significant. These results suggest that the subtest score items of KOSS partially mediate the relationship between grit and depression.

## 4. Discussion

To our knowledge, this is the first study to investigate the protective effects of passion and perseverance against depression and the mediating roles of occupational stress and perceived control in the relationship between passion and perseverance with depression among employees. Our findings demonstrate that high levels of passion and perseverance lead to lower occupational stress and a higher level of perceived control, thereby protecting against the onset of depression. We also demonstrated that among the various occupational stresses measured by KOSS, lack of reward had the greatest mediating effect between passion and perseverance with depression, whereas difficult physical environment had no significant mediating effect on either path. 

We found that occupational stress mediates the relationship between grit and depression. These results are consistent with previous findings [19,31] that high levels of grit negatively impact occupational stress. Previous studies also showed that when grit is high, recovery from negative emotions to positive emotions is faster even when receiving negative feedback [32], and psychological exhaustion is low [33], similar to our results. These results suggest that individuals with high levels of grit who have strong feelings toward long-term goals and who constantly challenge themselves, even in the face of adversity, feel relatively low stress even in adversity and can use it as an opportunity for growth [34]. A gritty individual seems to know how to cope (e.g., exhibits competence and ability to manage a given situation) in a stressful environment, which reduces stress, thereby resulting in less risk for depression.

Our exploratory investigation showed that high job demands, interpersonal conflict, job insecurity, organizational system injustice, lack of reward, and discomfort in occupational climate were individually partial mediators in the grit-depression relationship. In particular, among occupational stress, lack of reward showed the greatest mediating effect and difficult physical environment showed no significant effect on the grit-depression relationship. Insufficient job control had no significant mediating effect on the negative correlation between persistence and depression. This result is similar to that of a previous study [35] that examined the correlations and odds ratios between subscale values of KOSS and depression. There was no significant correlation between a difficult physical environment and depression in this study. However, unlike the results of this study, where lack of reward and high job demand act as the biggest predictors of depression, in previous studies, inadequate social support and discomfort in occupational climate showed the strongest relationships with increased risk of depression and was interpreted as a result of collectivism unique to Korea. This difference may be due to the higher proportion of highly educated subjects and the inclusion of more white-collar workers in this study than in previous studies. This difference is consistent with previous studies [36,37] in which white-collar workers reported significantly higher job demand stress and less interpersonal conflict stress than blue-collar workers. Blue-collar employees experience more behavioral stress, such as conflict with their supervisors, while white-collar employees experience more cognitive stress, such as excessive workload or high responsibility.

In mediating the relationship between passion and depression, insufficient job control of occupational stress had little effect compared to other types of stress, and there was no significant mediating effect of insufficient job control on the relationship between persistence and depression. However, perceived control was found to significantly mediate between both types of grit and depression. This suggests that the cognitive appraisal of controllably recognizing a stressful situation rather than the stress itself is important for predicting the occurrence of depression. This result is in line with those of previous studies [38,39] which show that rather than the stressor itself, individual cognitive appraisal of a stressor is more likely to be associated with symptoms of depression. A previous study confirmed that individuals with high perceived control levels experienced less psychological distress [40]. A previous meta-analysis also showed that people with high perceived control have lower depression severity [41], while a longitudinal study found that one month after experiencing a negative life event, the risk of developing a depressed mood is lower in people with high perceived control [42]. Wallston [43] defined perceived control as “the belief that one can determine one’s own internal states and behavior, influence one’s environment, and/or bring about desired outcomes”. Perceived control is a concept that reflects the internal locus of control, self-efficacy, mastery, and personal autonomy [40]. A high level of perceived control indicates a high level of belief that, in a given situation, the outcome is within the individual’s control. Such individuals are more active in using coping methods in stressful situations, which act as buffers against stress and increase the probability of achieving successful outcomes. The learned helplessness model holds that past uncontrollable outcome experiences will lead to cognitive and motivational deficits [44], and cause depression by giving people the belief that the future will also be uncontrollable [45]. Not all negative life events are expressed as depression, but a previous study [46] found that perceived control has a significant and central mediating effect through the learned helplessness model in experiencing depression. Another prior study showed that perceived control levels are lower for individuals of a lower level of socioeconomic class, who are less educated, and those who are unemployed [47]. As such, the social environment can affect perceived control level, which has a preventive effect on the occurrence of depression in employees, so it may be important to provide a social environment in which an individual can have a sufficient perceived control level.

This study had several limitations. First, since only participants in the workplace mental health examination at Kangbuk Samsung Hospital were included in the study, there may be limitations in generalizing this to all workers. However, our sample includes a variety of occupational groups, unlike many previous studies that focused on specific occupational groups. Second, because of its cross-sectional nature, longitudinal trajectories of changes in depressive symptoms could not be determined in this study. Specifically, the mediators could be covariates rather than mediators [48]. Thus, longitudinal studies are required to elucidate the relationships between variables and to extend the current findings. We collected data using self-reported measures, and thus, our findings may reflect response bias. However, careless responses were excluded during data processing. Future studies with objective measures could clarify the effects of grit on mental health.

## 5. Conclusion

Despite the limitations of the present study, this was the first investigation of potential mediators of the association between grit and depression in employees. These findings serve as a solid foundation for future studies designed to understand the specific mechanisms underlying grit and depression. Based on our data, forming high levels of grit in adolescence and early adulthood may be important for reducing stressful experiences in the work environment and for maintaining mental health. Along with perceived control, which showed a greater mediating effect on the relationship between grit and depression than the stress caused by insufficient job control, clinical intervention strategies for employees will be more effective when focusing on cognitive therapies aimed at modifying cognitive appraisals.

## Figures and Tables

**Figure 1 brainsci-13-00009-f001:**
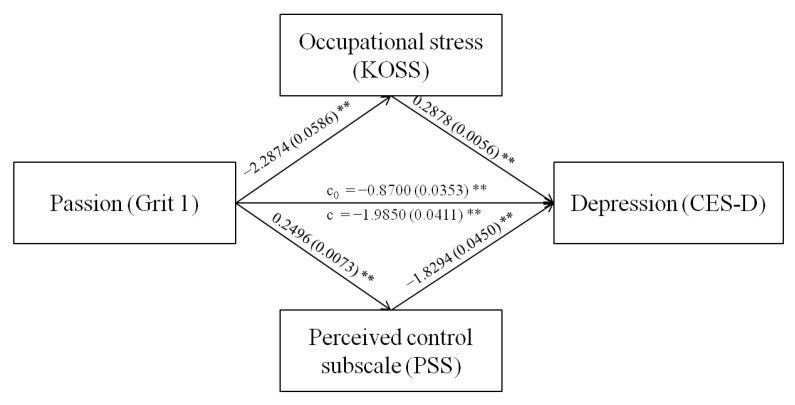
Mediation model of occupational stress and perceived control between passion and depression. The coefficient c is the total effect between passion and depression, and c0 is the direct effect of passion on depression while controlling for occupational stress and perceived control. ** *p* < 0.001.

**Figure 2 brainsci-13-00009-f002:**
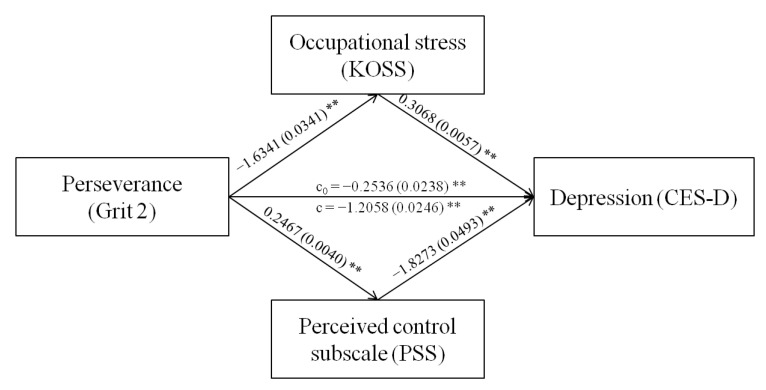
Mediation model of occupational stress and perceived control between perseverance and depression. The coefficient c is the total effect between perseverance and depression, and c0 is the direct effect of perseverance on depression while controlling for occupational stress and perceived control. ** *p* < 0.001.

**Table 1 brainsci-13-00009-t001:** Demographic and clinical characteristics of the study participants.

	Participants (n = 11,422)
Sex	
Male, n (%)	7750 (67.9)
Female, n (%)	3672 (32.1)
Age (years), mean ± SD	36.7 ± 9.4
Education (graduate)	
High school or below, n (%)	1404 (12.3)
College graduate, n (%)	1402 (12.3)
University graduate, n (%)	6841 (59.9)
Master’s degree, n (%)	1452 (12.7)
Doctorate degrees, n (%)	323 (2.8)
Marital status	
Married, n (%)	6369 (55.8)
Unmarried, n (%)	4808 (42.1)
Other, n (%)	245 (2.1)
Years of service (years), mean ± SD	10.5 ± 9.1
Hours of work per week (hours), mean ± SD	46.8 ± 7.5
Monthly earned income (million won), mean ± SD	4.3 ± 2.7
Grit passion subscale score, mean ± SD	9.79 ± 2.07
Grit perseverance subscale score, mean ± SD	17.28 ± 3.45
PSS control subscale score, mean ± SD	4.60 ± 1.68
KOSS total score, mean ± SD	40.82 ± 14.01
Difficult physical environment score, mean ± SD	5.75 ± 1.75
High job demand score, mean ± SD	10.31 ± 2.55
Insufficient job control score, mean ± SD	9.37 ± 2.18
Inadequate social support score, mean ± SD	6.05 ± 1.68
Job insecurity score, mean ± SD	3.71 ± 1.46
Organizational injustice score, mean ± SD	9.38 ± 2.40
Lack of reward score, mean ± SD	6.96 ± 1.87
Discomfort in occupational climate score, mean ± SD	8.46 ± 2.43
CES-D score, mean ± SD	14.49 ± 10.01

SD, standard deviation.

**Table 2 brainsci-13-00009-t002:** Inter-correlation matrix.

	Mean (Standard Deviation)	Grit Passion Subscale	Grit Perseverance Subscale	Occupational Stress	Perceived Control Subscale	Depression
Grit passion subscale	9.79 (2.07)	-				
Grit perseverance subscale	17.28 (3.45)	0.495 **	-			
Occupational stress	40.82 (14.01)	−0.378 **	−0.430 **	-		
Perceived control subscale	4.60 (1.68)	0.325 **	0.520 **	−0.443 **	-	
Depression	14.49 (10.01)	−0.440 **	−0.438 **	0.620 **	−0.548 **	-

** *p* < 0.001.

**Table 3 brainsci-13-00009-t003:** Results of multiple mediation analysis.

Regression Model		Goodness-of-Fit Indices			Regression Coefficient and Significance	
Outcome Variable	Predictor Variable	*R*	*R^2^*	*F*	*β*	*t*
Depression		0.47	0.22	397.96 **		
	Passion				−1.99	−48.34 **
Occupational stress		0.43	0.19	330.29 **		
	Passion				−2.29	−39.04 **
Perceived control		0.35	0.13	204.18 **		
	Passion				0.25	34.27 **
Depression		0.71	0.51	1176.25 **		
	Occupational stress				0.29	51.38 **
	Perceived control				−1.83	−40.61 **
	Passion				−0.87	−24.65 **
Depression		0.47	0.22	407.96 **		
	Perseverance				−1.21	−49.12 **
Occupational stress		0.48	0.23	434.74 **		
	Perseverance				−1.63	−47.89 **
Perceived control		0.53	0.28	554.84 **		
	Perseverance				0.25	62.30 **
Depression		0.70	0.49	1080.74 **		
	Occupational stress				0.31	53.64 **
	Perceived control				−1.83	−37.08 **
	Perseverance				−0.25	−10.64 **

** *p* < 0.01.

**Table 4 brainsci-13-00009-t004:** Bootstrap analysis of multiple mediation effects.

	Effect Size	SE	Percentage of Total Effect	95% CI	
				Lower Limit	Upper Limit
Passion → Depression					
Total effects	−1.9850	0.0411	100%	−2.0655	−1.9045
Direct effects	−0.8700	0.0353	43.83%	−0.9392	−0.8008
Total mediation effects	−1.1149	0.0315	56.17%	−1.1767	−1.0549
Passion → Occupational stress → Depression	−0.6582	0.0232	33.16%	−0.7043	−0.6132
Passion → Perceived control → Depression	−0.4567	0.0218	23.01%	−0.4998	−0.4144
Perseverance → Depression					
Total effects	−1.2058	0.0246	100%	−1.2539	−1.1577
Direct effects	−0.2536	0.0238	21.03%	−0.3003	−0.2069
Total mediation effects	−0.9522	0.0216	78.97%	−0.9955	−0.9094
Perseverance → Occupational stress → Depression	−0.5013	0.0158	41.57%	−0.5326	−0.4708
Perseverance → Perceived control → Depression	−0.4508	0.0169	37.39%	−0.4842	−0.4190

Confidence intervals, 95%; the number of bootstrap samples, 5000. SE, standard error; CI, confidence intervals.

## Data Availability

The data that would be necessary to interpret, replicate, and build upon the methods or findings reported in this article are available on request from the corresponding author S.J.C. The data are not publicly available because of ethical restrictions that protect patient privacy and consent.

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
