# Peer review of "Perceived Control and Work-Related Stress Mediate the Effects of Grit on Depression among Employees"

_brainsci, 2022, doi:10.3390/brainsci13010009_

Round 1

Reviewer 1 Report

Dear Author(s):

Great job! Just a few changes. On the first page, presentism (used multiple times) is not the correct word, it is presenteeism. Please check the entire document. On page 2, the last line in the first paragraph, there are grammar issues. In the first sentence of the second paragraph, what are the psychiatric disorders? The sentence beginning "As grit plays" does not make sense. Additionally, the second paragraph on page 2 does not make sense. The flow is very poor, there are many grammar issues and it is missing a significant amount of transitions. In section 2, the first three paragraphs must be deleted. In table 2, means and standard deviations need to be reported.

Author Response

Reviewer 1

Great job! Just a few changes.

Response:

Thank you for your valuable comments. We have revised our manuscript in response to your comments and provided point-by-point responses as below. We hope that you are satisfied with our revisions.

Comment #1

On the first page, presentism (used multiple times) is not the correct word, it is presenteeism. Please check the entire document.

Response:

We apologize for the typo errors. We have corrected the error in the Introduction as below:

Before revision:

(The 1st paragraph of INTRODUCTION)

(…) In the workplace, depression can negatively influence productivity through increased absenteeism and presentism. In South Korea, mean annual per person costs for absenteeism associated with depression were $181 and mean presentism costs per person associated with depression were $6,066.

After revision:

(The 1st paragraph of INTRODUCTION)

(…) In the workplace, depression can negatively influence productivity through increased absenteeism and presenteeism. In South Korea, the mean annual per person costs for absenteeism associated with depression were $181 and the mean presenteeism costs per person associated with depression were $6,066.

Comment #2

On page 2, the last line in the first paragraph, there are grammar issues.

Response:

We apologize for the grammar issues. We have corrected the error in the revised version of the manuscript as below.

Before revision:

(The 2nd paragraph in INTRODUCTION)

Despite the overlap in characteristics, individuals with high perseverance have an increased ability to maintain goals over long periods, unlike conscientious individuals who have high ability to complete tasks at hand.

After revision:

(The 2nd paragraph in INTRODUCTION)

Despite some similarities in characteristics, perseverance differs from conscientiousness. Although conscientious individuals complete tasks in one breath, individuals with high perseverance make steady and persistent efforts to achieve goals over long periods.

Comment #3

In the first sentence of the second paragraph, what are the psychiatric disorders?

Response:

Thank you for your thoughtful comments. In response to your comment, we have now presented the findings of previous research on the relationship between grit and symptoms of psychiatric disorders.

Before revision:

(The 3rd paragraph in INTRODUCTION)

Grit has been regarded as an important factor in coping with mental health pressure, and is associated with accompanying psychiatric disorders and greater health care management skills.

After revision:

(The 3rd paragraph in INTRODUCTION)

Since grit has been considered as an important factor in coping with mental health pressure, several previous studies have examined the relationship between grit and various psychiatric symptoms such as depression, anxiety, insomnia, and suicidal thoughts. In a cross-sectional study on university students in Thailand, gritty subjects showed lower levels of depression and anxiety. A study on emerging adults between the ages of 18 and 25 in the United States and a study on Turkish university students also showed that depression and anxiety were less likely to occur in subjects with high grit. Similarly, a study on the general population in Sweden during the COVID-19 pandemic showed that people with higher grit reported fewer symptoms of depression, anxiety, and insomnia during the pandemic era. In addition, a previous study showed that among individuals who experienced trauma, those with high grit experienced less posttraumatic stress symptoms and complained less about suicidal ideation.

Comment #4

The sentence beginning "As grit plays" does not make sense.

Response:

We appreciate your thoughtful comment. We have revised the sentence as follows:

Before revision:

(The 3rd paragraph in INTRODUCTION)

(…) As grit plays an important role in a variety of psychological factors, it is important for employees who are under stress but need to remain productive.

After revision:

(The 3rd paragraph in INTRODUCTION)

(…) Given that individuals with high grit experience fewer psychiatric symptoms and have more positive psychological factors, it is important to study grit in working place for employees who are inevitably exposed to stressful situations and need to maintain their mental health and productivity.

Comment #5
Additionally, the second paragraph on page 2 does not make sense. The flow is very poor, there are many grammar issues and it is missing a significant amount of transitions.

Response:

We appreciate your valuable critiques. We also apologize for the original manuscript lacking orderliness, which caused some confusion to the readers. As suggested in your comments, we have edited the second paragraph on page 2 as below. We would like to thank you for helping us improve the transition of our paper.

Before revision:

(The 3rd paragraph in INTRODUCTION)

Grit has been regarded as an important factor in coping with mental health pressure, and is associated with accompanying psychiatric disorders and greater health care management skills. A review paper found that grit showed positive associations with happiness, satisfaction and sense of belonging, psychological well-being, value and self-efficacy, self-esteem, growth mindset, pursuing engagement and pleasure in life, higher mental health, emotional stability during stressful or negative life events and sense of meaning in life and a negative correlation with perceived stress. Gritty individuals, who stay passionate and persevere against frustration, may interpret failure more optimistically and be less prone to negative emotions and to depressive episodes. Higher grit is negatively associated with maladaptive well-being outcomes. Although the relationship between grit and depression in individuals is generally consistent, the mechanisms by which grit protects against depression have not been fully elucidated. As grit plays an important role in a variety of psychological factors, it is important for employees who are under stress but need to remain productive. However, the number of studies examining the effect of grit among employees is insufficient.

After revision:

(The 3rd paragraph in INTRODUCTION)

Since grit has been considered as an important factor in coping with mental health pressure, several previous studies have examined the relationship between grit and various psychiatric symptoms such as depression, anxiety, insomnia, and suicidal thoughts. In a cross-sectional study on university students in Thailand, gritty subjects showed lower levels of depression and anxiety. A study on emerging adults between the ages of 18 and 25 in the United States and a study on Turkish university students also showed that depression and anxiety were less likely to occur in subjects with high grit. Similarly, a study on the general population in Sweden during the COVID-19 pandemic showed that people with higher grit reported fewer symptoms of depression, anxiety, and insomnia during the pandemic era. In addition, a previous study showed that among individuals who experienced trauma, those with high grit experienced less posttraumatic stress symptoms and complained less about suicidal ideation. There are preceding studies which show that grit plays an important role not only in various psychiatric symptoms, but also in various psychological factors that have recently been in the limelight, like positive psychological factors, quality of life, or happiness. Previous studies have shown that gritty individuals feel happier and have more life satisfaction and people with high grit have high self-efficacy, high self-esteem, low perceived stress, and are more emotionally stable during stressful or negative life events. Combining these previous studies, gritty individuals who stay passionate and persevere against frustration, may interpret failure more optimistically and be less prone to negative emotions and depressive episodes. Although the relationship between grit and depression in individuals is generally consistent, the mechanisms by which grit protects against depression have not been fully elucidated. Given that individuals with high grit experience fewer psychiatric symptoms and have more positive psychological factors, it is important to study grit in working place for employees who are inevitably exposed to stressful situations and need to maintain their mental health and productivity. However, the number of studies examining the effect of grit on employees is insufficient.

Comment #6

In section 2, the first three paragraphs must be deleted.

Response:

We apologize for the mistake. In response to your comment, we have removed those three paragraphs.

Comment #7

In table 2, means and standard deviations need to be reported. 

Response:

We appreciate your valuable suggestion. As suggested in your comments, we have included an edited table in our revised manuscript that presents the mean and standard deviations of variables. (Please refer to the revised Table 2).

Reviewer 2 Report

This manuscript has several strengths such as the sample size, however, the authors have not taken care of the details of the manuscript. E.g., the authors have forgotten part of the template. Please see:

"The Materials and Methods should be described with sufficient details to allow others to replicate and build on the published results. Please note that the publication of your manuscript implicates that you must make all materials, data, computer code, and protocols associated with the publication available to readers. Please disclose at the submission stage any restrictions on the availability of materials or information. New methods and protocols should be described in detail while well-established methods can be briefly described and appropriately cited."

It is not clear why individuals from 19 to 65 years old.

The instruments used should be cited in their validation in the target population.

The method should be better explained.

Minor

I would not specify that the SPSS PROCESS macro has been used in the abstract. Too specific.

Author Response

Reviewer 2

This manuscript has several strengths such as the sample size, however, the authors have not taken care of the details of the manuscript. E.g., the authors have forgotten part of the template. Please see:

"The Materials and Methods should be described with sufficient details to allow others to replicate and build on the published results. Please note that the publication of your manuscript implicates that you must make all materials, data, computer code, and protocols associated with the publication available to readers. Please disclose at the submission stage any restrictions on the availability of materials or information. New methods and protocols should be described in detail while well-established methods can be briefly described and appropriately cited."

Response:

We apologize for the lack of transparency of the original manuscript. We have thoroughly reviewed our drafts to make all the necessary changes. We hope that our efforts will meet your expectations.

Comment #1

It is not clear why individuals from 19 to 65 years old.

Response:

Only individuals aged over 19 years are included in this study, as they are regarded as adults who do not require parental consent according to legal standards. Individuals aged over 65 are excluded from the study participants in order to exclude the effect of neurocognitive changes on the results.

Before revision:

(The 1st paragraph in “2.1.participants” of METHOD)

The study participants were male and female workers ranging in age from 19 to 65 years who underwent workplace mental health screening at the Workplace Mental Health Institute of Kangbuk Samsung Hospital, Seoul, Republic of Korea, between April 2020 and March 2022.

After revision:

(The 1st paragraph in “2.1.participants” of METHOD)

The study participants were male and female workers who underwent workplace mental health screening at the Workplace Mental Health Institute of Kangbuk Samsung Hospital, Seoul, Republic of Korea, between April 2020 and March 2022. This study included participants over 19 at the beginning of the study in order to involve legal adults, and participants under 65 to prevent the effects of cognitive decline.

Comment #2

The instruments used should be cited in their validation in the target population.

Response:

We appreciate your valuable critiques. We added references for each assessment. Please refer to the changes in the revised METHOD.

Comment #3

The method should be better explained.

Response:

We appreciate your assistance in improving our manuscript. We have thoroughly reviewed our methods to make all the necessary changes. We hope that our efforts will meet your expectations. Please consider the correction as below:

Before revision:

(“2.3. Statistical analysis” of METHOD)

We conducted descriptive analyses of the demographic characteristics and correlation analyses of the four variables. Mediation analyses were performed using two independent variables (passion and perseverance), one dependent variable (depression), and two mediators (occupational stress and perceived control). Additional mediation analyses between passion and perseverance with depression were performed with subscale scores of KOSS as mediators. The mediation model was analyzed using Model 4 in the PROCESS Macro in SPSS, developed by Hayes [36]. For the best test of mediation effects, bootstrapping was carried out with 5000 samples to measure indirect effects and 95% confidence intervals were estimated. Control variables such as age, sex, education, marital status, years of service, hours of work per week, and monthly earned income were introduced in the model as covariates. If the confidence interval included zero, it meant that there was no significant mediating (indirect) effect at the significance level of 5%. All analyses were performed using SPSS 28.0 for Windows.

After revision:

(“2.3. Statistical analysis” of METHOD)

We conducted descriptive analyses of the demographic characteristics and correlation analyses of the four variables. To test the first and second hypotheses, Pearson correlations were performed for the scales to assess the direct associations between grit and depression and to support the associations between occupational stress and perceived control with the other variables, grit, and depression.

The mediation model was analyzed using Model 4 in the PROCESS Macro in SPSS, developed by Hayes. Bootstrapping is one of the most valid and powerful methods for testing multiple mediation models. Bootstrapping is a nonparametric resampling procedure which involves repeatedly sampling from the data set and estimating the indirect effect in each resampled data set. For the best test of mediation effects, bootstrapping was carried out with 5000 samples to measure indirect effects, and 95% confidence intervals were estimated. If the confidence interval included zero, it meant that there was no significant mediating (indirect) effect at the significance level of 5%. Mediation analyses were performed using two independent variables (passion and perseverance), one dependent variable (depression), and two mediators (occupational stress and perceived control). For each multiple mediation model, two specific indirect effects of grit on depression, one via occupational stress and the other via perceived control. A specific indirect effect represents the ability of occupational stress or perceived control to mediate the effect of grit on depression controlling for the other mediator. The total indirect effect of grit on depression summarizes the specific indirect effects. Additional mediation analyses between passion and perseverance with depression were performed with subscale scores of KOSS as mediators. Control variables such as age, sex, education, marital status, years of service, hours of work per week, and monthly earned income were introduced in the model as covariates. All analyses were performed using SPSS 28.0 for Windows.

Comment #4

I would not specify that the SPSS PROCESS macro has been used in the abstract. Too specific.

Response:

Thank you for your valuable comments. As you recommended, we have revised the expressions in the Abstract section to be less specific. The expression “SPSS PROCESS macro” has been replaced with “Mediation analysis”. We hope that the revised manuscript meets your requirements.

Before revision:

(ABSTRACT)

(…) SPSS PROCESS macro was used to determine relationships among trait-level passion and perseverance, work-related stress, perceived control, and depression.

After revision:

(ABSTRACT)

(…) Mediation analysis was performed to determine relationships among trait-level passion and perseverance, work-related stress, perceived control, and depression.

Round 2

Reviewer 2 Report

Thank you for addressing my comments